# Hardware Simulation for Analog Ultrasonic 2D Convolution

## Abstract

As its name suggests, the convolution operator is the basis and an essential component in Convolutional Neural Networks (CNNs). At the moment, modern CNN architectures rely heavily on parallel computation using GPUs and CPUs to perform many convolutions as fast as possible. However, the performance of computing CNNs is reaching its limit as the scaling of transistors approaches its size limits. The convolutional theorem suggests the possibility of using acoustic waves to efficiently perform the convolution operations through Fourier transforms in analog. This promises hardware that would be several orders of magnitude faster than existing silicon-based approaches. However, to date, nobody has shown the practical feasibility of such an approach. In this paper, we describe the first physics-based simulator for Ultrasonic Fourier Transform Convolutions (UFTC). By exploiting the diffraction nature of the waves, the Fourier transforms can be computed in the time it takes to propagate an ultrasonic wavefront. Our results show that ultrasonic computation could drastically improve the performance of CNNs by 12-458$\times$ FLOPS reduction and 1.3-4$\times$ computation speedup without loss of prediction accuracy.

## 1 Introduction

Computer vision, image processing, and pattern recognition have undergone several breakthroughs due to innovations in deep neural networks, computer architecture, and transistor scaling. These image-processing architectures are largely based on CNNs, which consists of many convolutional layers that extract features from images using trained kernels Li et al. (2022); Aloysius & Geetha (2017); Liu et al. (2017); Krizhevsky et al. (2012); Lecun et al. (1998); LeCun et al. (1989). One of the key reasons convolutional layers are so effective in these architectures is their inherent ability to be parallelized. Since convolutional layers operate by applying filters to small patches of an image independently, these operations can be distributed across multiple processing units, such as GPUs, CPUs, and FPGAs, to maximize computational efficiency.

Graphics Processing Units (GPUs) are one of the most widely used hardware for CNNs Dally et al. (2021); Ridnik et al. (2021); Zhu et al. (2020); Min et al. (2021). This is due to the number of cores and architecture that are suitable for convolutions and matrix multiplications, leading to deep learning libraries that support parallel processing through GPUs. However, it has become increasingly challenging to make advancements in speed and power for these processing units as the size of the transistor approaches nanometer scales. Furthermore, lowering power consumption and managing thermal dissipation have become more challenging, which are critical issues for some applications, such as Edge and mobile devices. Field Programmable Gate Arrays (FPGA) are another processing technology that is increasingly being used for CNN models and machine learning Ma et al. (2018); Mohammad & Agaian (2009); Bai et al. (2018); Ahmad & Pasha (2020); Bekkerman et al. (2012); Zhang et al. (2015). FPGAs are flexible and can be highly customized for parallel processing, enabling improved performance for CNN models compared to traditional GPUs and CPUs Russo et al. (2012); Li et al. (2016); Zhou & Jiang (2015); Wang et al. (2017); Podili et al. (2017). However, designing and optimizing FPGA systems can be time-consuming, complex, and costly. Furthermore, FPGAs have limited resources, such as memory and logic cells, to run massive CNN models, making it difficult to scale for larger CNN models.

One method to accelerate the convolution operation is to use the convolution theorem. Based on this theorem, the Fourier transforms of the input and the kernel can be point-wise multiplied, and the inverse Fourier transform can be applied to compute the convolution. When using the convolution theorem, the computation time for convolutions depends on the speed of the Fourier transform, the piece-wise multiplication, and the inverse Fourier transform. For the direct calculation of 2D Discrete Fourier Transform (DFT), the computation complexity is $\mathcal{O}((MN)^2)$ for an input of MxN images. The Fast Fourier Transform (FFT) is an efficient algorithm to compute the DFT and, using the Radix-2 FFT, the computation complexity becomes $\mathcal{O}(NMlog(NM))$. However, the FFT has a limited throughput for large data sets, even with parallelization. For this reason, this method is comparably faster only for larger kernels in modern GPUs.

An alternate method of using the convolution theorem is to use the wave phenomenon of diffraction to compute Fourier transforms. Optical wave transforms, for example, have been used in the past to compute the Fourier transform. An image is propagated in space and focused on a focal plane, generating the Fourier transform of the input image. Recently, ultrasonic waves have been used to realize Fourier transforms Hwang et al. (2022); Rippel et al. (2015); Liu et al. (2018b); Mathieu et al. (2014); Hwang et al. (2024), which can lead to a much more compact system owing to the lower speed of sound compared to the speed of light. The Ultrasonic Fourier Transform (UFT) hardware consists of Complementary Metal-Oxide Semiconductor (CMOS) components and AlN piezoelectric transmitters and receivers Nair et al. (2020); Chi et al. (2020); Pratt et al. (2017); Highlander & Rodriguez (2016). The ultrasonic Fourier hardware requires a phase shift module for the incoming waves to compute the Fourier transform and to reduce the device's size. One approach is to add a phase shifter on the transmitter and receiver sides. In another approach, a microfabricated convex acoustic lens is used as a phase shifter.

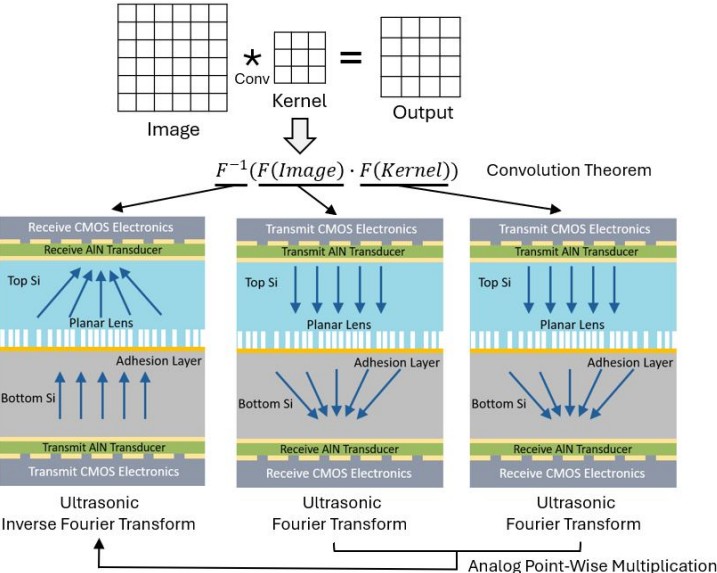

Figure 1: In the ultrasonic 2D convolution simulation, the convolution is computed by two ultrasonic Fourier transforms and one ultrasonic inverse Fourier transform. These simulations and models provide pathways to establish a hardware accelerator that would be several orders of magnitude faster than existing silicon-based approaches.

Based on the recent success in realizing the hardware components of the UFT, this paper explores the use of the UFT simulation to compute convolutions for CNNs. We use an UFT simulator based on the Huygens-Fresnel wave propagation to demonstrate and optimize the UFT hardware for CNNs. The Huygens-Fresnel is a fundamental principle used as a framework to analyze and predict the propagation of waves through space, including sound and light. The principle states that each point on a wavefront is the source of the secondary wavelet, which is combined to predict the propagation of the wave at any given time. Using this theorem, the simulation predicts the propagation of ultrasonic waves through different mediums, including silicon wafers. In order to simulate the ultrasonic

hardware accelerator, the input image is transmitted as an acoustic wave at the input. The transmitted wave passes through the first lens, placed at the focal length. After passing through the first lens, the ultrasonic wave generates the Fourier transform at the focal plane of the first lens. The same method is applied in parallel to the kernel after padding it to the same size array as the input image. The two Fourier transform outputs are multiplied in the analog domain, and the inverse Fourier transform is generated by propagating the product back to the kernel plane, resulting in a convolution operation, as is shown in Figure 2. We depict this procedure in Figure 2. This UFT convolution (UFTC) is applied for all convolution layers in the training model.

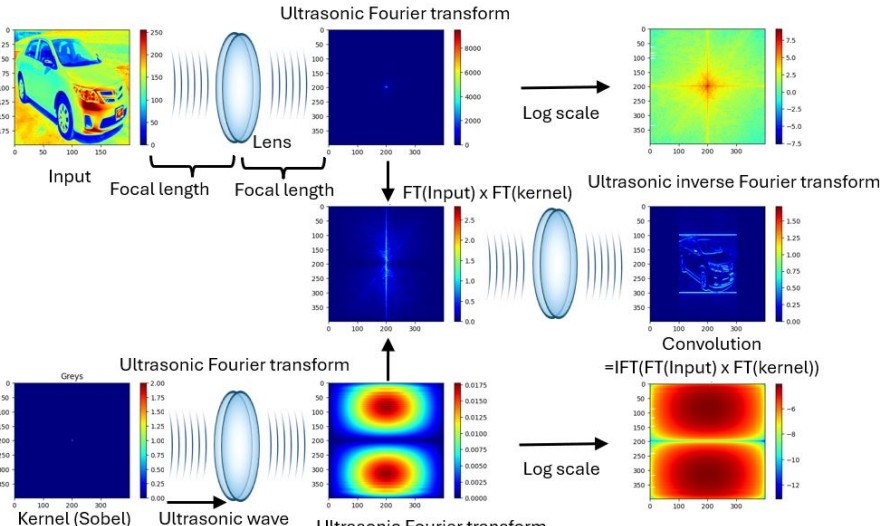

Figure 2: Convolution can be computed from two ultrasonic Fourier transforms and one inverse Fourier transform using the convolution theorem as shown in the picture. The input image and the kernel can run in parallel. Then, the two Fourier transforms are point-wise multiplied, and applied with inverse Fourier transform.

The total computation time for a single UFTC operation is twice the transit time of the wave between the transmitter and receiver, once for the Fourier transform and another for the inverse Fourier transform. When the focal length is 20mm, and the acoustic speed in silicon is 8,433 m/s, the convolution operation takes $2.37\mu$s. The analog multiplication of the image and kernel FT time can be neglected, considering the required time is around a few 10 ns.

**Contributions** With any new hardware research, it can be *extremely costly* to develop a fully scaled device. As a result, it is **crucial** to validate these theoretical frameworks through realistic simulations and models. Our current proposed hardware is still under development; however, several critical components of the technology were demonstrated in previous works, such as 2D arrays of transmitters and receivers of 1-2GHz, ultrasonic lenses, and working subsystems. For the first time, this paper presents a comprehensive, realistic simulation model in which an ultrasonic wave convolution tool has been used in practical CNN models. To our knowledge, we are one the first to demonstrate a physics-based wave simulation that can train various CNN models to accuracies similar to that of traditional CNNs while increasing its efficiency as determined by the speed of the ultrasonic waves. Just as the GPU technology has allowed various machine learning models to be accelerated, we hope our paper will excite researchers about the potential that this hardware may have in accelerating machine learning models. In short, the Deep Neural Network (DNN) community may benefit from having a new hardware component to optimize and perhaps create new architectures.

## 2 RELATED WORK

**Wave propagation simulation.** Wave simulation models compute the propagation of elastic pressure waves through different mediums in various physical systems. The simulation depends on factors such as the material properties, anisotropy, and accuracy required for the simulation.

For these simulations, different numerical techniques are used, such as the Finite Difference Time Domain (FDTD) method and Finite Element Method (FEM), which are widely used in industrial and open-source projects Wang & Xu (2015); Koene et al. (2017); Weedon & Rappaport (1997); Nagra & York (1998); Wang & Tripp (1996). These methods are accurate and have various supports for different materials and structures. However, it is hard to connect the physical outcome to CNN models to test the device as the core of a hardware accelerator. FDTD and FEM are resource-heavy when the mesh size is required to be only a few $\mu$m, and the 3D model is 10 cm long. Instead, we use ultrasonic wave simulation based on the Huygens-Fresnel principle to compute the far-field response at the focal planes.

**Ultrasonic sensors and Fourier hardware.** Continued research and developments in ultrasonic sensors have greatly enhanced the performance and applications of ultrasonic devices. Advances have led to miniaturized ultrasonic sensors that operate at GHz frequencies, enabling higher resolution and improved sensitivity Abdelmejeed et al. (2018); Baskota et al. (2022). In previous work Kuo et al. (2017), a CMOS-compatible aluminum nitride (AlN) transducer array running at 1.3GHz RF pulses was presented as an ultrasonic impedance imager. This ultrasonic sensor is estimated to have a 15-20 $\mu$m resolution. Furthermore, there has been research on optically measured and controlled Fourier transform using ultrasonic devices Liu et al. (2018a; 2019).

## 3 ULTRASONIC FOURIER TRANSFORM

When used with an ultrasonic lens, the ultrasonic wave generates the Fourier transform at the focal plane of the lens. This can be shown by ultrasonic wave simulation based on the Huygens-Fresnel principle. This simulation framework is important for several reasons. Firstly, the ultrasonic tool can verify and validate the acoustic Fourier transform hardware architecture. Finite Element Method (FEM) and Finite Difference Time Domain (FDTD) are alternate choices that can simulate the acoustic wave for both the longitudinal and shear waves using 3D meshes. However, it is nearly impossible to scale and simulate larger devices of 1920x1080 Full-HD pixels while maintaining a mesh size of a few $\mu$m or below, which is required by components such as acoustic lenses. In order to overcome these limitations, we have made a C++ and Python simulation framework to compute the ultrasonic wave at GHz frequencies. Secondly, the ultrasonic simulation can be used to optimize the structure of the device and fine-tune physical parameters such as the focal length, pixel size, acoustic wavelength, and input image width. These are important factors when designing the hardware to compute Fourier transform close to the exact FFT value. Thirdly, the program can easily be scaled and combined with machine learning libraries for training computer vision models.

### 3.1 THEORETICAL BACKGROUND

The propagation of elastic waves can be predicted by the Huygens-Fresnel principle Baker & Copson (2003); Berest & Veselov (1994); Pao & Varatharajulu (1976); Enders (1996). The Huygens-Fresnel's principle is derived from Green's function. The equation can be used to compute the wave propagation through different mediums and lenses for the far and near fields, defined as the Fresnel and Fraunhofer zones Goodman (2005). The necessary integrals are

$$U(P_1) \;=\; \frac{1}{j\lambda} \iint U(P_0) \frac{exp(jkr)}{r} cos\theta \, ds \tag{1}$$

where $U(P_1)$ and $U(P_0)$ are the magnitude of the field on the output plane and the input plane, $\lambda$ is the wavelength, k is the wavenumber, r is the distance between the source and the target, and $\theta$ is the angle between the normal direction and r, respectively.

### 3.2 ULTRASONIC FOURIER SIMULATION

In order to compute the Ultrasonic Fourier transform without approximations, we expand Equation 1 as

$$U(x,y) \;=\; -\frac{jz}{\lambda} \iint U(\alpha,\beta) \frac{\cos kr}{r^2} \, d\alpha \, d\beta + \frac{z}{\lambda} \iint U(\alpha,\beta) \frac{\sin kr}{r^2} \, d\alpha \, d\beta \tag{2}$$

where k is the angular wave number, $\lambda$ is the wavelength, r is the distance between the source and the target, x and y are the output coordinates, and $\alpha$ and $\beta$ are the input coordinates. The UFT was implemented, as shown in Figure 3, by directly calculating the integral for an input image to the FT plane. In order to compute the UFT such that its efficacy could be tested in the DNN models, a linear model of the UFT was developed. Here, the input 2D image plane is used to compute the 2D output plane by matrix-vector multiplication. In this operation, the UFT process was modeled by a $N^2 \times N^2$ matrix with $N^2$ input and output vectors, where the image is $N \times N$ pixels. The very large matrix-vector calculation could be computed using a GPU in a few seconds for $N = 128$.

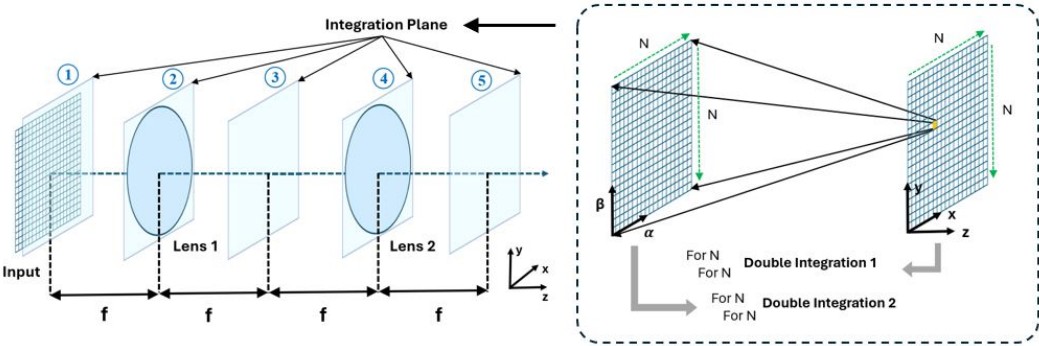

Figure 3: The ultrasonic wave simulation consists of two ideal convex lenses with a focal length of f. The first lens is used to generate the Fourier transform, and the second lens is used to validate the simulation.

Additionally, as shown in the previous section, the ultrasonic wave in the far field contains an extra quadratic phase term relative to the scaled Fourier transform. It is possible to digitally remove this phase term from the far field while still achieving a higher-speed Fourier transform than a GPU. However, this phase term can also be removed by introducing an acoustic lens placed at the transmitter's focal length, as shown in Figure 4. When the wave's distance is located exactly at the lens's focal length, the quadratic phase term disappears, leaving a scaled 2D Fourier transform.

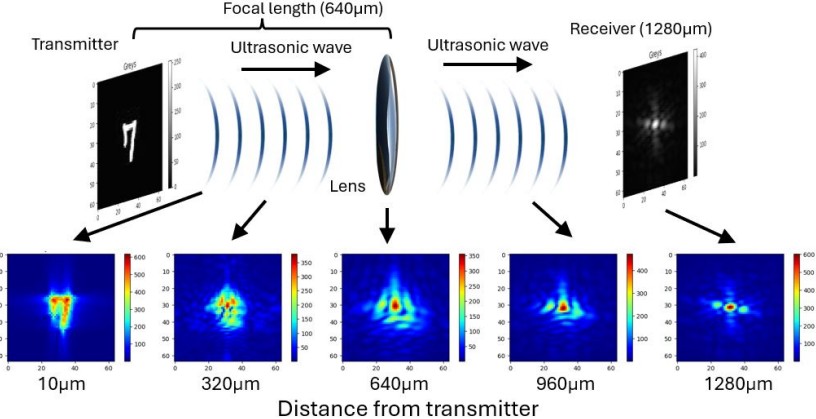

Figure 4: The ultrasonic wave propagation is used to compute the Fourier transform and convolution. The propagation of the input image is captured as it passes through a single lens and forms the Fourier transform.

### 3.3 ULTRASONIC CONVOLUTION

The convolution theorem states that

$$conv(\boldsymbol{x}, \boldsymbol{y}) = F^{-1}\left(F(\boldsymbol{x}) \cdot F(\boldsymbol{y})\right) \tag{3}$$

where $x$ and $y$ are inputs of the convolution operation, and $F$ represents the Fourier transform. This theorem allows convolution operations to be translated as mathematically equivalent Fourier transforms and element-wise multiplications.

As a member of the Fourier transform family, ultrasonic Fourier transforms can play the role of $F$ in Equation 3 and perform convolution operations. The corresponding computation graph is shown in Figure 2. More specifically, the ultrasonic Fourier transform is obtained by computing Equation 2 just before entering the lens, adding the phase term from the lens, and calculating the wave plane at the focal point as shown in Figure 2.

In convolutional neural networks, the computation cost scales linearly with the size of both inputs. For efficiency concerns, input images are usually rapidly downsampled by the first few convolutional/pooling layers, and convolutional kernels are often limited to $1 \times 1$ or $3 \times 3$ He et al. (2016); Huang et al. (2017). Larger convolutional kernels have been studied Ding et al. (2022), but scalability remains a non-neglectable concern and these large kernels have to be sparsified to reduce computation cost. Now, with ultrasonic Fourier transform, the computation time of convolutional operations is no longer dependent on kernel size. To convolve a $N \times N$ image with a $k \times k$ kernel, the computation cost can be drastically reduced from $\mathcal{O}(N^2 k^2)$ to $\mathcal{O}(N)$, enabling future research on gigantic convolutional kernels.

### 3.4 ULTRASONIC FOURIER TRANSFORM COMPUTATION TIME

The computation time of the ultrasonic 2D Fourier transform and the 2D FFT on a GPU is shown in this section. When the volume is limited to $127mm \times 50.8mm \times 40mm$ and the size of the single pixel transducer is $5\mu m$, 160 ultrasonic convolutions for a 1000x1000 input image can be performed in parallel at 2.4 $\mu s$. The computation time to run 160, 1000x1000 ultrasonic 2D Fourier transform is given by

$$T_{ultrasonic} = \frac{2 \cdot F}{c_p} \times 2 = \frac{2 \cdot 5mm}{8433m/s} \times 2$$
$$= 2.4\mu s = O(N) \tag{4}$$

where F is the focal length, and $c_p$ is the ultrasonic speed in silicon. In a RTX 4090 GPU with 82.6TFLOPS fully running in parallel, the time of computation for 160 2D FFT is

$$T_{GPU} = \frac{FLOP_{2DFFT} \times 160}{FLOPS_{GPU}} = \frac{5 \cdot N^2 Log N^2 \times 160}{82.6TFLOPS} = \frac{5 \cdot 1000^2 Log 1000^2 \times 160}{82.6TFLOPS} = 192\mu s \tag{5}$$

The loading of the input numbers and the reading of the output numbers will also require time, but that time would be required for any FFT calculation. Since the transit time of the ultrasonic array is $\frac{2F}{c_p}$, which is on the order of 1-4$\mu s$, once the aperture is transmitted, the new memory can be stored in the registers. Similarly, as the received input is being read out, the output values can be read out in $O(N^2)$ time of reading time steps, which might appear to be the bottleneck for calculations. In order to resolve this, parallel readout of each row of the 2D array will be implemented in CMOS. Each of the lines will be fed into a digitizer for amplitude and phase with multiplier factors stored in each multiplier. This CMOS multiplier array will be on the sides of each row. The parallel extraction and multiplications will reduce the reading and writing to an $O(N)$ operation.

### 3.5 ULTRASONIC FOURIER OPTIMIZATION

The dimension of the ultrasonic hardware needs to be optimized to generate a Fourier transform close to the exact FFT value. This is critical when using ultrasonic computation for CNN models to generate high-accuracy results. In order to avoid over and under-sampling, the dimension of the ultrasonic device needs to meet the critical sampling condition given by $L\Delta x = \lambda z$ , where L is the full width of the input, $\Delta$x is the pixel size of the transmitter, $\lambda$ is the wavelength of the acoustic wave, and z is the focal length, respectively Voelz (2011). In order to optimize the device, we first define the wavelength. Each pixel size should be designed to be substantially larger than the wavelength. Therefore, higher frequency and smaller wavelength are preferable to reduce the device's size. In

this work, we use frequencies between 1.0 and 2.0GHz, resulting in wavelengths between 8.4$\mu$m and 4.2$\mu$m for a longitudinal acoustic speed of 8,433 m/s in silicon. Secondly, the total number of pixels and the width of each pixel are defined. The pixel width is determined by the IC design and size of the CMOS technology. Pixel widths of 10-100$\mu$m are used in this simulation. Furthermore, if the total number of input pixels is 1000x1000 and the pixel width is 10$\mu$m, the entire width of the input, L, is 1mm. Finally, the focal length, z, is defined by the critical sampling equation. Using this technique, we have optimized the parameters of the wave simulation to achieve an ultrasonic Fourier transform close to the exact FFT, as shown in Figure 5.

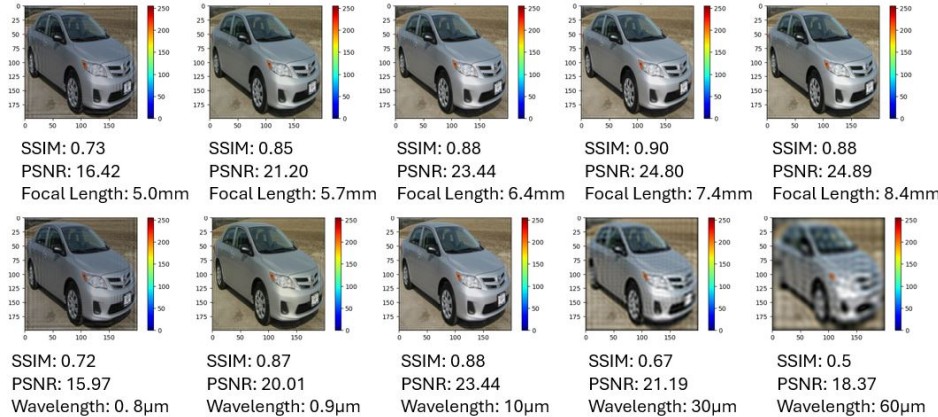

Figure 5: The dimension of the ultrasonic Fourier hardware needs to meet the critical sampling criteria. The focal length was swept around the critical point to assess the image quality using Structural Similarity (SSIM) and Peak Signal-to-Noise Ratio (PSNR) after passing through the Fourier and inverse Fourier transform. The default size of the padding, image resolution, image width, focal length, and wavelength were 100, 200, 1.6mm, 6.4mm, and 10 $\mu$m, respectively, as shown in the center image of each row.

## 4 HARDWARE FOR ULTRASONIC FOURIER TRANSFORM

The proposed ultrasonic Fourier hardware architecture includes a CMOS layer and AlN piezoelectric transmit and receive pixel arrays. In order to compute the convolution, the device simultaneously computes the input and kernel's Fourier transform, multiplies the results in the analog domain, and computes the inverse Fourier transform.

### 4.1 ULTRASONIC FOURIER ACCELERATOR AND HARDWARE

There have been many studies in ultrasonic transmitters and receivers, including applications for Fourier transform Liu et al. (2018a; 2019); Abdelmejeed et al. (2019b); Hoople et al. (2013); Abdelmejeed et al. (2019a). In previous work, the ultrasonic receiver and a fixed pattern transmitter were used to demonstrate wave propagation in silicon Hwang et al. (2024). As shown in Figure 6, the device consists of a piezoelectric MEMS structure and AlN transducers. The transmitted ultrasonic wave was measured electronically at 1.85GHz using a Geegah ultrasonic imager, which has 128x128 pixel arrays of 130nm CMOS-integrated 50x50$\mu$m AlN transducers.

### 4.2 POWER CONSUMPTION

The ultrasonic pixel consumes power through transmit and receive circuitry that is active only in a single UFT calculation. This power consumption was measured for a real 32x32 hardware chip during one convolution cycle. This measurement is split between key hardware blocks that allow for full convolution computation, in particular, the clock tree, aluminmum nitride(AlN) charging, variable gain amplifier (VGA), receive mixer, and output buffer. Here, the reported 130mA at 3.3V comes from static analog output buffers used for outputting ultrasonic pixel values. In contrast, the 640 mA at 1.2V is split between the clock tree and VGA which are the dominant power hungry devices. The

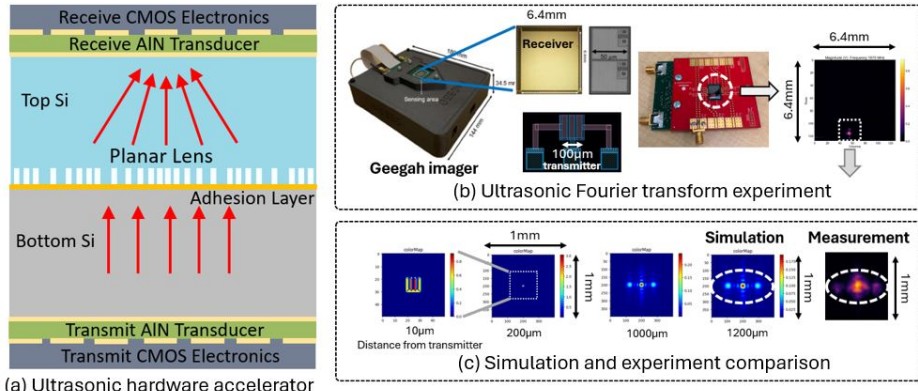

Figure 6: The architecture of the ultrasonic Fourier hardware for CNN is shown in (a). The actual hardware and preliminary results are shown in (b). The comparison between the simulation and the experiment is shown in (c) Hwang et al. (2024).

mixer and AlN consume 1.2V current but this draw is insignificant compared to the clock tree and VGA. The hardware remains consistent within every pixel which allows for ease of scaling for larger input arrays (2000x2000) at the cost of area and power. Performing this computation leads to an expected peak power consumption of 3630 W that is only active for 100 ns. Thus giving an energy consumption of $82.7 \mu J$ over the 100 ns period.

### 4.3 SCALABILITY

The size of a single pixel transducer can be reduced to below $5x5\mu m$. When the input image size is 1000x1000 pixels, then the size of a chip for a single convolution would be 5x5mm. We propose to assemble at least 160 ultrasonic chips in one device with a total volume of $127mm \times 50.8mm \times 40mm$ that could run 160, 1000x1000 convolutions in 2.4 $\mu s$ which is equivalent to 600 TFLOPS. The proposed device can be connected to a computer through PCIe, and the kernels will be transferred to the device's SRAM. The ultrasonic device will continuously run the convolution at the same speed, which can significantly reduce the CNN computation time in comparison with the conventional method.

## 5 EXPERIMENT RESULTS

### 5.1 IMPLEMENTATION DETAILS

**Datasets.** We consider four image classification datasets: MNIST Deng (2012), FashionM-NIST Xiao et al. (2017), CIFAR10 and CIFAR100 Krizhevsky et al. (2009). All 4 datasets include 50k training images and 10k testing images. Images in MNIST and FashionMNIST are of size $28 \times 28$ and in grayscale; images in CIFAR10 and CIFAR100 are of size $32 \times 32$ and colored. CIFAR100 includes 100 different classes while the other 3 datasets each includes 10.

**CNN Architectures.** We selected common convolutional neural network architectures, including LeNet LeCun et al. (1998), AlexNet Krizhevsky et al. (2017), DenseNet Huang et al. (2017), EfficientNet Tan & Le (2019), GoogLeNet Szegedy et al. (2015), MobileNet Howard et al. (2017), ResNet He et al. (2016), and VGG Simonyan & Zisserman (2014). We use the PyTorch Paszke et al. (2019) implementation of these networks, and create a Ultrasonic Fourier transform twin for each of these architecture following Equation 3.

**Image Classification Training.** We use the same training schedule for all model architectures and all datasets. Models are trained with batch size 128, for 25 epochs with cross-entropy loss Mao et al. (2023). The optimizer is SGD Ruder (2016) and the initial learning rate is 0.1, which decays in the 10th, 15th, and 20th epoch. All experiments are performed on a Nvidia A6000 GPU.

## 5.2 Computation Efficiency

The baseline method is the general convolution without any Fourier transforms running on CUDA GPU A6000, and the proposed implemented method is the convolution using ultrasonic diffraction and convolution theory. The FFT convolution was not used in this work but is expected to have a similar result as the general convolution.

In order to compare the efficiency, we use an internal function of Pytorch to track and calculate the total FLOPs. For example, the MNIST LeNet with and without ultrasonic convolution changed from 290.6K to 50.6K FLOPS in Table 2. The A6000 GPU used 290.6k FLOPS to compute the LeNet using the general convolution and only 50.6k FLOPS with ultrasonic diffraction. The difference of 240k (290.6k-50.6k) FLOPS was assigned to the ultrasonic diffraction convolution and is assumed to be outside of the A6000 GPU calculation. Also, the FLOPS reduction was calculated by dividing 290.6k by 50.6k. Additionally, the proposed fully scaled ultrasonic convolution has a computation power of 600 TFLOPS, and therefore, the 240k FLOPS computed in the ultrasonic device were assumed to be very fast.

The computation efficiency between original convolutional architectures and their ultrasonic Fourier transform twins are shown in Table 1. We observe $12\text{-}458\times$ FLOPS reduction and $1.3\text{-}4\times$ computation speedup on the majority of common models achieved by UFTC.

Table 1: Computational cost with and without applying UFTC. We report float point operations (FLOPS) and computation time (batch size 256) of popular convolutional neural networks in standard ImageNet training, *i.e.* taking inputs of shape $3 \times 224 \times 224$.

| Model | Sonic Conv | FLOPS ↓ | Computation Time / ms ↓ | Model | Sonic Conv | FLOPS ↓ | Computation Time / ms ↓ |
|---|---|---|---|---|---|---|---|
| AlexNet | ✗ | 714.2M | 12.00 | MobileNet_V3_Large | ✗ | 226.0M | 26.00 |
| | ✓ | **58.6M** | **3.65** | | ✓ | **12.0M** | **22.47** |
| DenseNet121 | ✗ | 2.9B | 100.99 | ResNet18 | ✗ | 1.8B | 33.25 |
| | ✓ | **32.4M** | **77.48** | | ✓ | **5.5M** | **12.94** |
| DenseNet161 | ✗ | 7.8B | 196.76 | ResNet34 | ✗ | 3.7B | 49.49 |
| | ✓ | **61.1M** | **140.36** | | ✓ | **8.0M** | **22.52** |
| DenseNet169 | ✗ | 3.4B | 122.80 | ResNet50 | ✗ | 4.1B | 80.66 |
| | ✓ | **39.8M** | **97.99** | | ✓ | **24.4M** | **50.42** |
| DenseNet201 | ✗ | 4.3B | 157.63 | ResNet101 | ✗ | 7.8B | 126.52 |
| | ✓ | **51.4M** | **126.27** | | ✓ | **34.6M** | **77.71** |
| EfficientNet_V2_S | ✗ | 2.9B | 89.55 | ResNet152 | ✗ | 11.6B | 178.00 |
| | ✓ | **29.0M** | **66.03** | | ✓ | **47.3M** | **110.33** |
| EfficientNet_V2_M | ✗ | 5.4B | 150.68 | VGG11 | ✗ | 7.6B | 76.48 |
| | ✓ | **46.4M** | **105.45** | | ✓ | **123.7M** | **34.47** |
| EfficientNet_V2_L | ✗ | 12.3B | 263.75 | VGG13 | ✗ | 11.3B | 125.75 |
| | ✓ | **80.5M** | **175.36** | | ✓ | **123.7M** | **49.68** |
| GoogLeNet | ✗ | 1.5B | 36.76 | VGG16 | ✗ | 15.5B | 148.00 |
| | ✓ | **7.5M** | **26.58** | | ✓ | **123.7M** | **53.86** |
| MobileNet_V3_Small | ✗ | 59.6M | **9.54** | VGG19 | ✗ | 19.6B | 166.09 |
| | ✓ | **4.8M** | 15.21 | | ✓ | **123.7M** | **58.18** |

## 5.3 Performance on Image Classification

Due to computation constraints, we are only able to fit the UFTC simulation for 4 architectures into the GPU: LeNet, ResNet18, ResNet34, and DenseNet121. We compare the classification accuracy of trained models in Table 2. At a cost of 0.4%-25.7% performance drop, UFTC is able to reduce computational cost by 12-458 times.

## 6 Limitations and Discussions

The ultrasonic wave consists of longitudinal and shear waves. The Huygens-Fresnel principle used in this work focuses on the longitudinal waves, and the final result needs to be scaled for comparison with experimental results Cohen (1967). The amplifier signal-to-noise ratio (SNR) will determine the effective number of bits (ENOB) of the FFT digitization and may affect the accuracy of the system. From the perspective of future hardware development, the computation time and power consumption of the ultrasonic convolution greatly depend on the feature size of the CMOS technology. Building the ultrasonic transmit and receive chips using a smaller CMOS technology is advantageous for higher

Table 2: Classification performance w/wo applying UFTC. We report float point operations (FLOPS), computation time (batch size 4096) and classification accuracy of popular convolutional neural networks on common image classification datasets.

| Dataset | Model | Sonic Conv | FLOPS ↓ | Computation Time / ms ↓ | Accuracy / % ↑ |
|---|---|---|---|---|---|
| MNIST | LeNet | ✗ | 290.6K | 1.74 | **99.42** |
| | | ✓ | **50.6K** | **0.67** | 98.95 |
| | ResNet18 | ✗ | 33.1M | 10.10 | **99.54** |
| | | ✓ | **91.6K** | **6.27** | 98.78 |
| | ResNet34 | ✗ | 69.8M | 18.71 | **99.40** |
| | | ✓ | **139.0K** | **11.18** | 98.95 |
| | DenseNet121 | ✗ | 56.9M | 62.42 | **99.57** |
| | | ✓ | **650.8K** | **47.30** | 98.57 |
| Fashion MNIST | LeNet | ✗ | 290.6K | 1.74 | **90.75** |
| | | ✓ | **50.6K** | **0.67** | 89.04 |
| | ResNet18 | ✗ | 33.1M | 10.10 | **92.41** |
| | | ✓ | **91.6K** | **6.27** | 89.43 |
| | ResNet34 | ✗ | 69.8M | 18.71 | **91.87** |
| | | ✓ | **139.0K** | **11.18** | 89.58 |
| | DenseNet121 | ✗ | 56.9M | 62.42 | **92.84** |
| | | ✓ | **650.8K** | **47.30** | 88.43 |
| CIFAR10 | LeNet | ✗ | 664.3K | 2.81 | **69.38** |
| | | ✓ | **71.5K** | **0.85** | 60.67 |
| | ResNet18 | ✗ | 37.1M | 12.01 | **78.72** |
| | | ✓ | **107.0K** | **6.70** | 65.19 |
| | ResNet34 | ✗ | 74.9M | 25.00 | **77.49** |
| | | ✓ | **158.2K** | **11.69** | 64.10 |
| | DenseNet121 | ✗ | 58.5M | 63.25 | **81.28** |
| | | ✓ | **650.8K** | **47.29** | 57.86 |
| CIFAR100 | LeNet | ✗ | 671.9K | 2.82 | **35.80** |
| | | ✓ | **79.1K** | **0.85** | 26.97 |
| | ResNet18 | ✗ | 37.2M | 12.00 | **50.33** |
| | | ✓ | **153.1K** | **6.70** | 24.66 |
| | ResNet34 | ✗ | 75.0M | 21.06 | **47.76** |
| | | ✓ | **204.3K** | **11.69** | 24.95 |

transistor density, improved performance, and lower power. However, there are challenges, such as Integrated Circuit (IC) layout complexity, manufacturing precision, and higher cost of advanced CMOS node fabrication.

# 7 CONCLUSION

In this work, we presented a simulation framework for an ultrasonic device that can compute convolutions in CNN models with 12-458× FLOPS reduction and 1.3-4× speedup through the Fourier transform using the intrinsic characteristics of ultrasonic waves in CMOS/piezoelectric arrays. A simulation model was first used to optimize the model's dimensions, including the wavelength, focal length, pixel size, and width of the input image. Then, the optimized hardware simulation model was used to train CNN models, which consistently showed high accuracy in predictions for several DNN models. This model demonstrates that there is a pathway to greatly reduce the time and energy consumption of CNNs using ultrasonic physical convolution accelerator systems.

ACKNOWLEDGMENTS

We would like to acknowledge the Defense Advanced Research Projects Agency (DARPA), ACCESS program, and the NSF program for supporting this work. We would also like to acknowledge the Cornell Nanoscale Facility (CNF), which is part of the National Nanotechnology Coordinated Infrastructure (NNCI) and the National Science Foundation (Grant ECCS-1542081).

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

## A APPENDIX

## S1 THEORETICAL BACKGROUND

Huygens-Fresnel principle is the fundamental equation used to predict the propagation of wave Baker & Copson (2003); Berest & Veselov (1994); Pao & Varatharajulu (1976); Enders (1996). The integration equation without approximations is given by Goodman (2005)

$$U(P_1) = \frac{1}{j\lambda} \iint U(P_0) \frac{exp(jkr)}{r} cos\theta \, ds \tag{6}$$

The propagation of the wave can be further expanded by substituting

$$cos\theta = \frac{z}{r} \tag{7}$$

After substituting the $\cos\theta$ term, the equation becomes

$$U(x,y) = \frac{z}{j\lambda} \iint U(\alpha\beta) \frac{exp(jkr)}{r^2} \, d\alpha \, d\beta \tag{8}$$

which can be used to compute the propagation of waves without any approximations. In this work, the integral without approximation is used to simulate the convolution neural network. However, for analytical purposes, the original integral can be further approximated by replacing the denominator as

$$r = \sqrt{z^2 + (x-\alpha)^2 + (y-\beta)^2} = z\sqrt{1 + \left(\frac{x-\alpha}{z}\right)^2 + \left(\frac{y-\beta}{z}\right)^2} \tag{9}$$

The binomial expansion can be approximated as

$$(1+x)^\alpha \approx 1 + ax + \frac{a}{2}(a-1)x^2 \tag{10}$$

The square root term can be approximated as

$$\sqrt{1+x} \approx 1 + \frac{x}{2} - \frac{x^2}{8} \tag{11}$$

Using the binomial expansion of the square root term, we get

$$r = z\left(1 + \frac{1}{2}\left(\frac{x-\alpha}{z}\right)^2 + \frac{1}{2}\left(\frac{y-\beta}{z}\right)^2\right) \tag{12}$$

The $r$ term can be replaced to get the Fresnel diffraction integral in the near field as

$$U(x,y) = \frac{1}{j\lambda z} \iint U(\alpha,\beta) \, e^{jkz\left[1 + \frac{1}{2}\left(\frac{x-\alpha}{z}\right)^2 + \frac{1}{2}\left(\frac{y-\beta}{z}\right)^2\right]} \, d\alpha \, d\beta \tag{13}$$

The $r^2$ term in the denominator is approximated as z while dropping all the other terms. However, the other expansion terms are significantly large for the exponential value and are retained. Rearranging, the final wave equation becomes

$$U(x,y) = \frac{e^{jkz}}{j\lambda z}e^{j\frac{k}{2z}(x^2+y^2)}\iint \left[U(\alpha,\beta)e^{j\frac{k}{2z}(\alpha^2+\beta^2)}\right]e^{-j\frac{2\pi}{\lambda z}(x\alpha+y\beta)}\,d\alpha\,d\beta \tag{14}$$

When the wave is near the source, it has additional terms to the 2D Fourier transform. In order to get the 2D Fourier transform from the wave, the equation is further approximated when the wave is far from the source in the Fraunhofer region. This is when z is large enough as

$$z \gg \frac{k(\alpha^2+\beta^2)_{max}}{2} \tag{15}$$

Given these approximations, the wave equation becomes

$$U(x,y) = \frac{e^{jkz}}{j\lambda z}e^{j\frac{k}{2z}(x^2+y^2)}\iint U(\alpha,\beta)\,e^{-j\frac{2\pi}{\lambda z}(x\alpha+y\beta)}\,d\alpha\,d\beta \tag{16}$$

The terms inside the double integral can be substituted by the following to compare with the exact 2D Fourier transform.

$$u = \frac{\alpha}{\lambda z} \quad , \quad v = \frac{\beta}{\lambda z} \tag{17}$$

Except for the exponential-of-quadratic term outside the integral, the double integral can be compared to the 2D Fourier transform given by

$$U(x,y) = \iint U(u,v)\,e^{-j2\pi(xu+yv)}\,du\,dv \tag{18}$$

The additional quadratic term in front of the integral for the far field can be eliminated by introducing a lens in front of the transmitter, as shown in the next section.

## S2    ULTRASONIC FOURIER SIMULATION

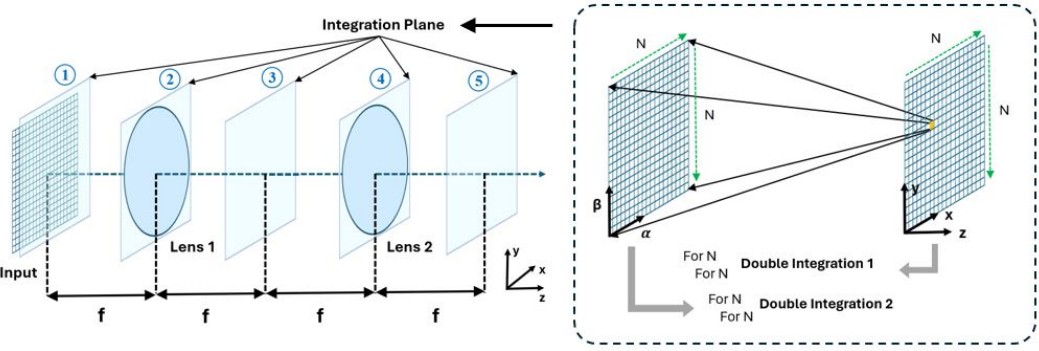

Figure S7: The ultrasonic wave simulation consists of two ideal convex lenses with a focal length of f. The first lens is used to generate the Fourier transform, and the second lens is used to validate the simulation.

As shown in the previous section, the ultrasonic wave in the far field contains an extra quadratic phase term to the scaled Fourier transform. It is possible to digitally remove these phase terms from the far field while still achieving a higher-speed Fourier transform than a GPU. However, these phase terms can also be removed by introducing an acoustic lens placed at the transmitter's focal length. The propagation of the ultrasonic wave with the lens placed at a given position d is given by

$$U(x,y) \;=\; \frac{e^{j\frac{k}{2f}\left(1-\frac{d}{f}\right)\left(x^2+y^2\right)}}{j\lambda f} \times \iint U_{input}(\alpha,\beta)\, e^{-j\frac{2\pi}{\lambda f}(x\alpha+y\beta)}\, d\alpha\, d\beta \tag{19}$$

Where k is the angular wave number, $\lambda$ is the wavelength, and f is the focal length, respectively. When the wave's distance is located exactly at the lens's focal length, the quadratic phase term disappears, leaving a scaled 2D Fourier transform. This is one way to design the Fourier hardware. It is also possible to remove the lens and add phase shifters on the transmitter and receiver sides. This would result in extra calculations but would still yield a faster CNN than current GPUs.

