# OpenReview forum: "Hardware Simulation for Analog Ultrasonic 2D Convolution"
_ICLR.cc/2025/Conference — ICLR 2025 Conference Withdrawn Submission_

### Official Review · Reviewer_5grG · 2024-10-30

**Soundness:** 2
**Presentation:** 1
**Contribution:** 2
**Rating:** 3
**Confidence:** 2

**Summary:**

The paper aims to provide insights into potential hardware for accelerating CNNs based on an ultrasonic hardware-based Fourier transform. The main contribution of the study is to demonstrate the effectiveness of such hardware using simulation. The following points summarize the findings:
* Convolution can be computed by using a Fourier transform, inverse Fourier transform, and dot-wise multiplication operation.
* Building hardware for performing Ultrasonic Fourier Transform Convolutions (UFTC) in the analog domain can calculate Fourier and inverse Fourier transforms with low power and higher speed.
* Creating a simulator as a proof of concept for such hardware, which can evaluate the impact on end-to-end performance of CNN models, is highly desirable.
* The paper describes the details of such a simulator and presents results on multiple CNN architectures.

**Strengths:**

* The evaluation of new hardware techniques aimed at accelerating neural networks is a highly important research direction.
* The paper explores the novel idea of using ultrasonic waves for Fourier-based convolution.

**Weaknesses:**

* The paper is not written coherently and lacks clarity about contribution. Please clarify the points added to "Questions" sections. It is also be advisable to revisit the points to make paper more readable.
* the paper presents simulator as the major contribution. However, the description of simulator and its validation is missing. It will be helpful to build confidence in simulator's efficacy to represent the real hardware.

**Questions:**

* Line 218-220 mentions "The UFT wasimplemented, as shown in Figure 3, by directly calculating the integral for an input image to the FT plane. In order to compute the UFT such that its efficacy could be tested in the DNN models, a linear model of the UFT was developed."
   > What is the impact of using a linear model of UFT? Does it preserve the accuracy and performance of the model?

* Line 241-246: What is the significance of removing the quadratic term? Does the simulation system assume the lenses are placed at the focal length? This point is not clear from the text.

* Section 3.4: The section introduces several specific numerical values. Are these values used as examples for demonstration, or do they hold specific significance? For example, is the size of the single pixel transducer, 5 μm, critical?

* Section 3.4: In the expression, FLOP2DFFT is replaced with 5.N^2 log N^2 without explanation. Please provide clarification.

* Line 441-446 : "The difference of 240k (290.6k-50.6k) FLOPS was assigned to the ultrasonic diffraction convolution and is assumed to be outside of the A6000 GPU calculation. Also, the FLOPS reduction was calculated by dividing 290.6k by 50.6k."
  > Please share rationale regarding the assumptions.

---

### Official Review · Reviewer_Tmh6 · 2024-10-31

**Soundness:** 3
**Presentation:** 3
**Contribution:** 3
**Rating:** 3
**Confidence:** 4

**Summary:**

The paper introduces a Hardware Simulation for Analog Ultrasonic 2D Convolution, proposing an ultrasonic Fourier transform-based hardware architecture to accelerate CNN computations. By simulating an analog ultrasonic wave propagation system, the authors aim to perform convolutions via Fourier transforms in an analog format, potentially reducing computational complexity and energy consumption compared to traditional digital methods. The approach leverages the convolution theorem and ultrasonic wave diffraction to execute Fourier transforms in the analog domain. Results from the hardware simulation show a potential FLOPS reduction of 12-458× and a computation speedup of 1.3-4×, demonstrating competitive accuracy for common CNN models.

**Strengths:**

The proposed approach presents a novel application of ultrasonic wave physics to CNNs, offering an energy-efficient alternative to digital accelerators. By leveraging the convolution theorem and performing Fourier transforms in the analog domain, the method reduces computational load and could provide significant benefits for applications requiring low-power, high-speed computing.

The experimental results are compelling, with up to 458× FLOPS reduction and competitive CNN accuracy across popular models like ResNet and DenseNet. This demonstrates the method’s potential in practical CNN applications, especially where energy efficiency is critical, such as edge and mobile devices.

The paper makes a valuable interdisciplinary connection between ultrasonic hardware and deep learning, which could inspire future research in analog computation for machine learning. The authors also present a realistic hardware simulation model, addressing practical parameters like focal length and pixel size, which strengthens the relevance and feasibility of the proposed system.

**Weaknesses:**

The method’s reliance on specific hardware parameters, such as ultrasonic wave speed and focal length, might limit its adaptability across different types of hardware or more complex neural network architectures. Further investigation into the system's flexibility could enhance its applicability.

Although the performance gains are promising, the use of analog hardware raises potential concerns about scalability and integration with current digital infrastructures. Additional discussion on interfacing this approach with digital processing units could provide more clarity on its practical deployment.

The accuracy drop noted in some models suggests that the ultrasonic convolution method may need further optimization for complex CNN architectures. Clear guidelines on tuning for different CNN structures could help improve performance consistency across a wider range of applications.

**Questions:**

Could the authors discuss potential methods for integrating the ultrasonic analog architecture with digital processing units to create a hybrid system?

Given the noted accuracy drop in more complex models, would it be feasible to develop a specialized tuning procedure to enhance performance consistency?

Since the system’s design is reliant on specific hardware parameters, what challenges might arise if this method were adapted for different hardware setups?

---

### Official Review · Reviewer_wk4z · 2024-11-04

**Soundness:** 3
**Presentation:** 3
**Contribution:** 3
**Rating:** 5
**Confidence:** 3

**Summary:**

This paper presents a novel approach to accelerating Convolutional Neural Networks (CNNs) by leveraging ultrasonic waves for convolution operations. This work explores the use of ultrasonic Fourier Transform Convolutions (UFTC), a method that replaces digital convolution operations with an analog Fourier transform computed by propagating ultrasonic waves. A physics-based simulator for this purpose is introduced, showing significant improvements in FLOPS reduction (up to 458×) and computation speedup (1.3-4×) compared to conventional methods.

**Strengths:**

Contribution: The paper proposes an innovative analog computing solution that offers substantial computational benefits, addressing fundamental limitations in current silicon-based architectures.

Simulation-Based Validation: This work demonstrates the feasibility of UFTC using a well-constructed simulation, which serves as an essential step toward physical implementation.

Detailed Theoretical Framework: The authors thoroughly explain the theoretical underpinnings of their approach, particularly how ultrasonic waves can compute Fourier transforms and convolve inputs with minimal latency.

Outcome: The reported reduction in computation time and FLOPS demonstrates the potential for large-scale deployment, especially in resource-constrained environments.

**Weaknesses:**

My main two concerns are listed below:

Limited Hardware Implementation: While the simulation results are promising, the actual hardware demonstration may differ from simulations, which might lead to performance discrepancies in real-world applications.

Accuracy Trade-offs: The accuracy reduction (up to 25.7% on certain datasets) raises questions about the generalizability of UFTC, particularly for applications requiring high precision.

**Questions:**

There is no clear explanation why there is a significant accuracy drop in some cases in Table 2? Please explain.

---

### Official Review · Reviewer_ZQHf · 2024-11-05

**Soundness:** 3
**Presentation:** 3
**Contribution:** 2
**Rating:** 3
**Confidence:** 4

**Summary:**

The author proposes a simulation framework for an ultrasonic device that can compute convolutions in CNN models with 12-458x FLOPS reduction and 1.3-4x speedup through the Fourier transform using the intrinsic characteristics of ultrasonic waves in CMOS/piezoelectric arrays.

**Strengths:**

Strength:
1.	It explores an ultra-sonic implementation of Fourier-domain convolution and demonstrates some FLOPS reduction and speedup.

**Weaknesses:**

Weaknesses

1.	How is the complex number in the Fourier domain handled in the real hardware?

2.	Hardware nonideality is not considered in the modeling and evaluation. There is significant difference between simulation and measurement as shown in Fig 6.

3.	The peak power of the proposed system is 3630W, which raises concerns about practicality.

4.	The novelty is limited. Fourier-domain convolution is not invented here. Using wave diffraction, e.g., 4-f optical system, for convolution is a well explored literature.

5.	The accuracy drop is huge and unacceptable in the application demonstrated. It is hard to justify why this method is practically useful with such a large degradation.

6.	The paper claims a simulation framework for UFTC. What is new in this framework? Any special kernel implementation and optimization to speed up the training?

**Questions:**

Weaknesses

1.	How is the complex number in the Fourier domain handled in the real hardware?

2.	Hardware nonideality is not considered in the modeling and evaluation. There is significant difference between simulation and measurement as shown in Fig 6.

3.	The peak power of the proposed system is 3630W, which raises concerns about practicality.

4.	The novelty is limited. Fourier-domain convolution is not invented here. Using wave diffraction, e.g., 4-f optical system, for convolution is a well explored literature.

5.	The accuracy drop is huge and unacceptable in the application demonstrated. It is hard to justify why this method is practically useful with such a large degradation.

6.	The paper claims a simulation framework for UFTC. What is new in this framework? Any special kernel implementation and optimization to speed up the training?

---

### Note · Authors · 2025-01-22

I have read and agree with the venue's withdrawal policy on behalf of myself and my co-authors.